# Performance Analysis of Data Augmentation Approaches for Improving Wrist-Based Fall Detection System

**DOI:** 10.3390/s25072168

**Published:** 2025-03-29

**Authors:** Yu-Chen Tu, Che-Yu Lin, Chien-Pin Liu, Chia-Tai Chan

**Affiliations:** Department of Biomedical Engineering, National Yang Ming Chiao Tung University, Taipei City 112, Taiwan; sammymoontu@gmail.com (Y.-C.T.); jlcy025@gmail.com (C.-Y.L.); henry062439.be09@nycu.edu.tw (C.-P.L.)

**Keywords:** data augmentation, deep learning technology, wrist-based fall detection, wearable sensor

## Abstract

The aging of society is a global concern nowadays. Falls and fall-related injuries can influence the elderly’s daily living, including physical damage, psychological effects, and financial problems. A reliable fall detection system can trigger an alert immediately when a fall event happens to reduce the adverse effects of falls. Notably, the wrist-based fall detection system provides the most acceptable placement for the elderly; however, the performance is the worst due to the complicated hand movement modeling. Many works recently implemented deep learning technology on wrist-based fall detection systems to address the worst, but class imbalance and data scarcity issues occur. In this study, we analyze different data augmentation methodologies to enhance the performance of wrist-based fall detection systems using deep learning technology. Based on the results, the conditional diffusion model is an ideal data augmentation approach, which improves the F1 score by 6.58% when trained with only 25% of the actual data, and the synthetic data maintains a high quality.

## 1. Introduction

The health issues are worthy of notice in an aged society [1,2,3]. There were statistically about 506 million people over 65 in 2008, which will grow to an estimated 1.3 billion by 2040 [4]. In addition, according to the 2021 Current Population Survey’s Annual Social and Economic Supplement, reported by the U.S. Census Bureau, about 28% of U.S. older adults live alone [5]. Falls and fall-related injuries are the most critical health issues for older adults, especially for those living alone. In the United States, around 25% of older adults report a fall, and 10% of fall events lead to a serious injury every year [6]. Moreover, compared with older adults who live alone and with others, such as their family or health care professionals, there is a 6% chance of a higher incidence of falls over time [7]. The consequences of falls include physical damage, psychological effects, and financial problems. Physical damages, such as fractures and bruises [8,9], may reduce functional independence and have more chance of becoming recurrent fallers, even death. As to psychological effects, fear of falling again makes older adults lose confidence in their safety, which may cause a decline in their willingness to engage in activities of daily living (ADLs) [10]. As regards the financial problems, the overall medical cost for fatal and nonfatal falls was approximately $50 billion every year, resulting in substantial burdens for society [11].

In recent years, plenty of studies have been putting effort into developing an effective fall detection system to minimize the negative influence of falls. The aim of the fall detection system is to send an immediate alert when falls occur. Implementing one or multiple inertial measurement units (IMUs) in fall detection systems is the most common technique for acquiring daily motion information. An IMU consists of a tri-axial accelerometer, a gyroscope, and a magnetometer. The accelerometer provides the acceleration reference when moving, the magnetometer provides a heading reference, and the gyroscope measures how the orientation changes with time [12,13,14]. Chen et al. [15] used an IMU device worn on the waist to detect fall events. The threshold-based method was applied to determine the impact phase of falls. Also, considering the possibility that older adults could not report their self-situation when the falls occurred, a network of fixed motes in the home environment was used to target the location of victims. Hussain et al. [16] used the SisFall dataset [17], a public dataset that acquires motion data from an IMU worn on subjects’ waists, to accomplish an activity-aware fall detection system. The machine learning-based method was employed to detect the occurrences of falls and to recognize the ADLs before the falls. The result of the system using the k-nearest neighbors (KNNs) classifier achieved the highest 99.80% accuracy and the best 96.82% accuracy in recognizing multiple falling activities using random forest (RF).

The placement of the IMU also plays a vital role in developing a functional fall detection system [18,19,20,21]. Several details should be considered comprehensively, such as the performance and the acceptance. Chai et al. [18] evaluated the performance of a fall detection system using different sensor fusions. The best 94.10% accuracy occurs when the sensor fusion of IMUs worn on the chest, elbows, wrists, thighs, and ankles, a total of nine IMUs. As for the single IMU, the highest 92.51% accuracy occurs when the sensor is worn on the chest, and for the sensors worn on the wrists, however, the result has the worst 78.83% accuracy. Kangas et al. [19] aimed to assess different low-complexity fall detection algorithms using a tri-axial accelerometer attached at the waist, wrist, and head. The results suggested that the sensor placement should be at the waist or head to obtain the most sensitive fall detection. Although it is reasonable that the ideal position must be located at the trunk due to the usual stability associated with maintaining the upper body balance [22], users might not widely accept it because of discomfort. Hence, a wrist-based fall detection system seems the most acceptable for older adults owing to its user-friendliness and less stigma of using a medical device [23,24], but the worst performance should be solved. The reason for the worst performance of the wrist-based fall detection system is its high complexity motion modeling [25,26]. Hence, many researchers have focused on implementing deep learning technology into wrist-based fall detection systems due to the ability to extract features automatically [27,28]. However, fall events are rare in our daily lives, indicating that the deep learning-based fall detection system has a data imbalance issue, which may result in bias and potentially weaken the performance of deep learning technology in detecting falls.

This study aims to analyze the performance of different data augmentation approaches for a wrist-based fall detection system, including data transformation, synthetic minority over-sampling technique, autoencoder, variational autoencoder, and denoising diffusion probabilistic model. The UP-Fall Detection Dataset [29] is employed to validate different approaches to data augmentation since it provides a useful resource to fairly compare different data augmentation methods. The wrist-based IMU signals, especially the fall event signals, are expanded through the data augmentation algorithm. In addition, the influence on the performance of detection and the quality of synthetic data, such as the Train-on-Synthetic-Test-on-Real (TSTR) score and divergence comparison, is discussed in this paper.

## 2. Related Work

Five classic data augmentation algorithms are utilized to expand the volume of training data. In this section, the data augmentation algorithms are described briefly.

### 2.1. Data Transformation

Data transformation is the most straightforward data augmentation methodology. By tuning the physical properties of IMU signals, including rotation, permutation, time-warping, magnitude-warping, jittering, and scaling [30], data transformation can synthetically generate a large quantity of training data, expecting to solve the class imbalance issue, as shown in Figure 1.

### 2.2. Synthetic Minority Over-Sampling Technique (SMOTE)

Synthetic Minority Over-sampling Technique (SMOTE) [31] is a common approach to synthesizing new samples based on the distribution of minority classes. The SMOTE process has three steps to expand the data volume. Firstly, each sample in the minority class computes its k nearest neighbors using the Euclidean distance, commonly k = 3. Second, one sample is randomly chosen from these nearest neighbors, and new samples are generated following Equation (1):(1)$$X_{new}=X_{chosen}+\left|X_{nearest}-X_{chosen}\right|\times \delta ;\;\delta \in [0,\;1],$$
where $X_{chosen}$ denotes the initially selected sample, and $X_{nearest}$ denotes the sample selected from k nearest neighbors. Finally, all samples in the minority class will be chosen, and the above steps will be repeated to expand the minority class.

### 2.3. Autoencoder (AE)

Autoencoder (AE) [32] is an unsupervised deep learning algorithm. Typically, an autoencoder model is composed of two neural network models, namely, an encoder and a decoder. Firstly, the encoder reduces the dimension of the input data into latent vectors, which is a compressed, low-dimensional representation of the input data. Then, the decoder reconstructs the data based on the latent vectors as the output data. Finally, the AE model will minimize the error between input and output data to generate a clean signal.

### 2.4. Variational Autoencoder (VAE)

Variational Autoencoder (VAE) [33] is an advanced model of AE, which also consists of an encoder and a decoder. Unlike AE, a VAE model limits the encoding stage, making the latent vectors follow the Gaussian distribution. Additionally, because its mean value and standard deviation can parameterize the Gaussian distribution, we can theoretically generate whatever we want through the VAE model.

### 2.5. Denoising Diffusion Probabilistic Model (DDPM)

Denoising Diffusion Probabilistic Model (DDPM), or Diffusion Model [34], is a novel generative model. Two main processes are designed to generate samples based on training data: the forward diffusion process and the reverse diffusion process. First, the forward diffusion process is defined as a Markov chain; namely, the training data corrupt the Gaussian noise iteratively during the forward diffusion stage. Notably, when the iteration step is close to infinity, the noisy data can be seen as an isotropic Gaussian distribution. Second, the aim of the reverse diffusion process is to train a denoising model for denoising the latent vectors of Gaussian distribution to clean data iteratively.

## 3. Materials and Methods

This section provides the proposed validation method for different data augmentation approaches. It begins with a description of the dataset used in this study and the preprocessing techniques applied. The details of various data augmentation methods used for comparison are presented, outlining their implementation. Finally, the effectiveness of the proposed augmentation approach is assessed by evaluating its impact on fall detection performance using commonly applied evaluation metrics, which are discussed at the end of this section.

### 3.1. Dataset

In this study, the UP-Fall Detection Dataset [29] is adopted to evaluate the effectiveness of different data augmentation techniques. The dataset includes six types of activities of daily living (ADLs) and five distinct types of falls, with each activity being performed three times. It was collected from 17 healthy young participants, nine males and eight females, aged between 18 and 24 years. Data acquisition was conducted using a multimodal approach, incorporating wearable sensors, ambient sensors, and vision-based equipment, as illustrated in Figure 2b.

Specifically, five Mbientlab MetaSensor (published by Mbientlab, San Francisco, CA, USA) wearable devices were used to capture raw motion data from a three-axis accelerometer and a three-axis gyroscope at a sampling frequency of 18.4 Hz. These sensors were placed on the left wrist, under the neck, inside the right pants pocket, at the center of the waist (attached to a belt), and on the left ankle, as shown in Figure 2a. Only data collected from the sensor on the left wrist were utilized in this study to enhance user convenience in the daily use of the wrist-based fall detection system.

### 3.2. Data Preprocessing

To prepare the dataset for analysis, missing data were first removed. Specifically, trials 2 and 3 for subject 8 in activity 11 were excluded due to unknown equipment errors. After data cleaning, a sliding window technique was applied for segmentation. Fall samples are divided into 100 reading windows (approximately 5.43 s) with a 50% overlap to preserve temporal continuity. For ADL events, a different preprocessing approach was employed based on sample duration. If an ADL event sample contains fewer than 1000 readings (approximately 54.35 s), it is processed using the same sliding window technique as the fall samples. For longer ADL recordings, the event is divided into ten equal segments, and 100 reading windows are randomly extracted from each segment to ensure balanced data representation. After segmentation, the dataset contained a total of 2282 ADL windows and 756 fall windows. To normalize the input features, all segmented windows underwent min–max normalization, scaling the values to a range between −1 and 1.

### 3.3. Implementation and Configuration

To address the data imbalance issue, each data augmentation method is specifically designed to generate a greater number of synthetic fall data than ADL data. This approach ensures that, when the real and synthetic datasets are combined, the number of fall data matches that of ADL, thereby achieving a balanced distribution for the training dataset. Furthermore, to maintain fairness in the evaluation process, all augmentation methods generate several synthetic data equal to the size of the real training dataset. The following section details the implementation of different data augmentation methods.

First, three common data transformation techniques—rotation, scaling, and permutation—are employed to augment both ADL and fall signals. In the rotation process, the three-axis acceleration and the three-axis angular velocity signals are rotated independently in a randomly chosen direction. For scaling, the signals are either expanded or compressed by a randomly selected factor ranging from 0 to 2, adjusting the signal magnitude accordingly. The permutation transformation involves segmenting each signal into four equal-length parts and randomly rearranging their positions to introduce variability. These transformations are applied separately to generate augmented versions of the signals and are individually evaluated to assess their impact on the fall detection system. Finally, the results obtained from the three different transformation methods are averaged to determine their overall effectiveness in enhancing data augmentation.

As for the SMOTE, the number of nearest neighbors k is set to 5 as the default value, and the random state is fixed at 42 to ensure reproducibility of SMOTE. To maintain a consistent number of synthetic samples, the sampling strategy is designed to match the total number of synthetic data with that of the real data. Additionally, this approach ensures that the combined dataset of real and synthetic samples maintains an equal distribution between fall and ADL data.

The autoencoder, composed of an encoder and a decoder, is illustrated in Figure 3. The encoder compresses input signals into a latent space representation that preserves essential information. It consists of three convolutional layers (Conv1D) with 1 × 3 filters, each followed by a max-pooling layer (MaxPool1D) with a 1 × 2 kernel, creating an alternating structure. The decoder mirrors this arrangement with three transposed convolutional layers (Transposed Conv1D) and three up-sampling layers (Upsample), ensuring a balanced and symmetric architecture. Its objective is to reconstruct the original signals while retaining key features from the encoded representation.

The detailed architecture is presented in Table 1, where the numerical values correspond to the parameters of different layers: Conv1D (number of output channels, kernel size), MaxPool1D (kernel size), Transposed Conv1D (number of output channels, kernel size), and Upsample (size). The model is trained using the Adam optimizer with a learning rate of 3 × 10^−4^ to maintain stable learning dynamics. The mean squared error (MSE) is adopted as the loss function, with a batch size of 64, and training is conducted for a total of 100 epochs. The hyperparameters listed in Table 2 represent the ranges and values explored during the tuning process to identify the optimal configuration for the model.

The VAE follows an encoder–decoder structure and introduces probability and randomness, allowing the generation of new samples within a continuous latent space. This characteristic makes it well suited for generative modeling tasks. This architecture is illustrated in Figure 4 and detailed in Table 3.

In this framework, the encoder compresses input data into a latent distribution, learning to represent it using a mean (μ) and standard deviation (σ). It consists of four convolutional layers, each with 64 filters, followed by a max-pooling layer with a pooling factor of 2. At the bottleneck, two fully connected layers transform the latent features into the mean and standard deviation, defining a probabilistic latent space. The decoder starts with an upsampling layer of size 100, followed by four convolutional layers that reconstruct the data from the latent representation. The model is trained using the Adam optimizer with a learning rate of 5 × 10^−4^ and a batch size of 64. A customized loss function, combining mean squared error (MSE) and Kullback–Leibler (KL) divergence, is applied to balance reconstruction accuracy and latent space regularization. The model is trained for a total of 100 epochs. Table 4 presents the hyperparameter ranges designed and explored during tuning to determine the most effective model configuration.

To implement and configure DDPM, a simplified U-Net [35] structure is employed to enhance the diffusion model for data augmentation. U-Net is a widely used architecture in diffusion models due to its effective skip connections, which allow feature maps from corresponding encoder layers to be directly copied and concatenated into the decoder. This mechanism helps retain spatial details that might be lost during decoding, thereby improving the quality of generated data. Additionally, U-Net restores extracted features to their original dimensions through an up-sampling process, enabling the model to capture fine details and enhance the accuracy and diversity of generated samples.

Due to these advantages, U-Net has been extensively applied in various fields, including data synthesis [36] and medical image segmentation [37]. However, its original design is primarily designed for high-dimensional image processing in computer vision.

Whereas wearable sensor data are comparatively lower in dimensionality and follows a simpler structure. To better accommodate the characteristics of inertial sensor data, we adopt a modified U-Net architecture specifically designed for time-series data with shorter timestamps and fewer channels, as illustrated in Figure 5.

The proposed structure consists of both a down-sampling and an up-sampling process. The down-sampling process incorporates a convolutional layer (Conv1D), a self-attention mechanism, and a 1×2 max-pooling layer to progressively extract features. In the up-sampling process, we employ skip connections to restore lost information and consists of a Conv1D, a self-attention mechanism, and an up-sampling layer with a scaling factor of 2. Each Conv1D layer comprises a 1×3 convolutional kernel, followed by group normalization and a Gaussian Error Linear Unit (GELU) activation function, ensuring stable and efficient learning. The model is trained with 1000 diffusion timesteps, utilizing the Adam optimizer with a learning rate of 3 × 10^−4^. Training is performed with a batch size of 64 over 300 epochs. Table 5 outlines the hyperparameter search space considered in this study for identifying the most suitable model setup.

### 3.4. Evaluation Methodology

In this study, three evaluation metrics are employed to assess the effectiveness of data augmentation: fall detection performance, Train on Synthetic Test on Real (TSTR) score, and divergence comparison ratio. These metrics evaluate the impact of synthetic data augmentation on classification performance, the model’s ability to generalize from synthetic to real data, and the degree of similarity between synthetic and real data.

#### 3.4.1. Fall Detection Performance

To assess the effectiveness of the diffusion model-based data augmentation method in fall detection, synthetic data are integrated with real training data to train the fall detection model. The model architecture, as shown in Figure 6, consists of three convolutional blocks, followed by two fully connected layers and a softmax activation function. Each convolutional block includes a one-dimensional convolutional layer (Conv1D) with 64 filters of size 1 × 3, batch normalization (BN) to stabilize training, a rectified linear unit (ReLU) activation function for non-linearity, and max pooling (MP) with a 1 × 2 filter to reduce spatial dimensions. Additionally, a dropout layer is incorporated to mitigate overfitting and improve generalization.

During training, the Adam optimizer is employed with cross-entropy as the loss function. The learning rate is set to 3 × 10^−4^, and the model is trained using a batch size of 64 for 300 epochs. To evaluate its performance, four commonly used metrics—accuracy, precision, recall, and F1 score—are used to quantify the model’s ability to distinguish between fall events and ADL.(2)$$Accuracy=\frac{TP+TN}{TP+FP+FN+TN},$$(3)$$Precision=\frac{TP}{TP+FP},$$(4)$$Recall=\frac{TP}{TP+FN},$$(5)$$F1-score=2\times \frac{Recall\times Precision}{Recall+Precision}.$$

To assess the model’s ability to generalize to unseen data, the leave-one-group-out cross-validation (LOGO-CV) method was employed. This validation approach simulates a realistic scenario where the model is tested on subjects whose data are not included in the training set. The dataset, consisting of 17 subjects, is divided into four groups: three groups containing four subjects each and one group containing five subjects. During training, data from three groups are used to train the model, while the remaining group is held out for testing. This process is repeated until all groups have been used for testing, ensuring a robust evaluation of the model’s performance across different subjects.

The Train-on-Synthetic-Test-on-Real (TSTR) score is used to evaluate the effectiveness of synthetic data in fall detection tasks, specifically, its ability to differentiate between ADL and fall events. This metric assesses how well a model trained exclusively on synthetic data can generalize to real-world data. To compute the TSTR score, a fall detection model is first trained using only synthetic data and subsequently tested on real data. The model architecture and parameters follow the design specifications outlined in Section 3.4.1. Additionally, the leave-one-group-out cross-validation (LOGO-CV) method is applied to ensure a fair and unbiased evaluation, preventing data leakage across the training and testing phases.

A higher TSTR score indicates that the synthetic data closely resemble real data, effectively enabling the deep learning model to generalize well to real-world scenarios. This suggests that the synthetic data capture essential features necessary for fall detection. However, an excessively high TSTR score may indicate a lack of diversity in the generated data, potentially leading to model overfitting. Striking a balance between data similarity and diversity is crucial to ensuring the robustness and generalizability of the fall detection model.

#### 3.4.2. Divergence Comparison

The divergence comparison aims to assess the relationship between real and synthetic data, as well as the distinction between ADL and fall events within the synthetic data. To evaluate the quality of the generated data, inner-class and outer-class dispersion measures were employed, which quantify the distribution of data within and between classes.

Inner-class dispersion measures the degree of variation within each class by calculating the sum of squared deviations of individual data points from their respective class mean. The total inner-class dispersion matrix is obtained as a weighted sum of the dispersion matrices of all classes. Mathematically, it is represented as(6)$$S_{W}=\sum_{i=1}^{M}P(\Omega_{i})S_{W}^{\left(i\right)}\;=\sum_{i=1}^{M}P(\Omega_{i})\frac{1}{N_{i}}\sum_{k=1}^{N_{i}}\left(X_{k}^{\left(i\right)}-m^{\left(i\right)}\right){\left(X_{k}^{\left(i\right)}-m^{\left(i\right)}\right)}^{T}$$
where $S_{W}^{\left(i\right)}$ denotes the covariance matrix of the class $\Omega_{i}$, and $m^{\left(i\right)}$ represents the mean of that class. A lower inner-class dispersion indicates that the data points within a class are more tightly clustered around their mean, suggesting higher consistency and quality in the generated samples.

On the other hand, outer-class dispersion quantifies the degree of separation between different classes by measuring the average squared distance between their means. This metric captures how distinct the classes are in the feature space. It is defined as:(7)$$S_{B}=\frac{1}{2}\sum_{i=1}^{M}P(\Omega_{i})\sum_{j=1}^{M}P(\Omega_{j})S_{B}^{\left(ij\right)}\;=\frac{1}{2}\sum_{i=1}^{M}P(\Omega_{i})\sum_{j=1}^{M}P(\Omega_{j})(m^{\left(i\right)}-m^{\left(j\right)}){\left(m^{\left(i\right)}-m^{\left(j\right)}\right)}^{T}$$

A higher outer-class dispersion value indicates greater separation between classes, suggesting that the generated data preserve clear distinctions between different activity types.

To further analyze data quality, a ratio is computed to assess the relationship between two different groups of samples. The ratio is defined as(8)$$ratio=\frac{S_{B}}{S_{W}}$$

In this study, two key relationships are examined. The first considers the distinction between fall and ADL signals within the synthetic dataset. A higher ratio in this context indicates greater class separability, suggesting improved differentiation between ADL and fall signals. The second relationship evaluates the similarity between real and synthetic data. For this analysis, an equal number of samples from each class, both real and synthetic, are randomly selected. A lower ratio in this case suggests that the synthetic data closely resemble the real data, reflecting higher realism in the generated samples.

## 4. Results and Discussions

Table 6 presents the optimized hyperparameters that were explored during the tuning process to determine the most effective configurations for the autoencoder, VAE, and diffusion model. The selection of these hyperparameters was based on iterative experimentation aimed at achieving the best balance between model performance and training stability. All experimental results and performance improvements discussed in this study were obtained using data augmentation methods implemented with these optimized hyperparameter configurations.

### 4.1. Impact of Data Augmentation on Fall Detection Performance

The performance evaluation of the fall detection system, trained with various data augmentation techniques, is presented in Table 7. The evaluation is conducted using both the complete dataset and a reduced dataset containing only 25% of the real training data to simulate the scarcity of data in the real world. The process involved using real data to apply different data augmentation methods, implementing the fall detection classifier, and testing the system to obtain the final results.

The diffusion model-based data augmentation demonstrated the most significant improvement in fall detection performance among all methods. When trained on the full dataset, this method achieved the highest accuracy of 93.30% and an F1 score of 86.00%, outperforming other augmentation techniques. Even in the case where only 25% of the real training data are available, the diffusion model continued to yield superior results, attaining an accuracy of 91.75% and an F1 score of 83.28%. Compared with the baseline method (BL), which relies solely on real data without any augmentation, the diffusion model led to notable improvements. Specifically, it increases the F1 score by 2.13% when trained on the full dataset and by 6.58% when trained with only 25% of the real data.

This notable improvement can be attributed to the ability of diffusion models to generate high-quality, diverse synthetic samples. By leveraging probabilistic modeling, diffusion models learn the underlying data distribution. Through an iterative process of progressively adding and removing noise, they generate diverse and realistic synthetic samples that enhance the training set. This diversity enables the classifier to learn more generalizable and discriminative features, leading to improved performance, especially when real data are limited. These results highlight the strong potential of diffusion models for augmenting sensor data in fall detection applications.

### 4.2. Quality Assessment of Synthetic Data

The effectiveness of a fall detection system depends not only on the quantity of training data but also on their quality. Therefore, evaluating the quality of synthetic data generated by the diffusion model is essential. In this section, three key criteria are used to assess data quality: visualization of synthetic sensor signals, the Train-on-Synthetic-Test-on-Real (TSTR) score, and divergence analysis.

#### 4.2.1. Visualization of Sensor Signals

Figure 7 and Figure 8 display the accelerometer and gyroscope signals for real (left) and synthetic (right) data generated by the diffusion model for ADL and fall events, respectively. Each plot represents a three-axis sensor with 100 readings captured over a time window of approximately 5.43 s.

The diffusion model successfully captures and reproduces important signal characteristics of different activities. For instance, ADL activities such as walking exhibit periodic patterns reflected in the generated synthetic signals. Similarly, a key feature of fall events is the sudden impact, causing a sharp peak in the sensor readings. To illustrate this characteristic, the highest acceleration value in Figure 8 is marked with an “×.” The diffusion model effectively learns this pattern and generates synthetic signals that resemble real fall events.

#### 4.2.2. Train on Synthetic Test on Real Score

Table 8 presents the TSTR evaluation results for various data augmentation methods. Data transformation (DTF) and SMOTE exhibit the highest performance, indicating that both methods generate synthetic data that resemble real-world sensor signals. However, the exceptionally high similarity between synthetic and real data suggests a potential risk of overfitting, which may limit the model’s ability to generalize beyond the training dataset. As shown in Table 4, the fall detection performance with DTF and SMOTE shows slight improvement compared with the baseline, but they do not achieve the best results among all methods. In contrast, autoencoder (AE) and VAE yield significantly lower performance, suggesting that the synthetic data generated by AE and VAE may not sufficiently capture the underlying characteristics of real sensor signals, leading to suboptimal performance in distinguishing between ADL and fall events.

The diffusion model demonstrates a balance between synthetic data fidelity and diversity, achieving an accuracy of 91.30% and an F1 score of 84.67%. While its performance is slightly lower than that of DTF and SMOTE, it offers a significant advantage in mitigating overfitting by generating more diverse synthetic samples. This ability to generate diverse yet realistic samples enhance the model’s capacity to generalize across different activity scenarios, making it a promising approach for improving fall detection performance in real-world applications.

#### 4.2.3. Divergence Comparison on Synthetic Data

Table 9 presents a comparison of divergence between synthetic data for ADL and fall events, as well as between real and synthetic data generated by different data augmentation methods. Regarding the divergence between fall and ADL synthetic signals, the diffusion model exhibited an inner-class divergence of 0.0022 and an outer-class divergence of 0.00009, resulting in a ratio of 0.0400. This ratio is the highest among all data augmentation methods evaluated. A higher ratio indicates a more significant distinction between the two activity classes, suggesting that synthetic fall data generated by the diffusion model are significantly different from synthetic ADL data. This distinct separation allows the fall detection system to effectively learn the differences between these activities, ultimately enhancing its ability to classify falls accurately.

In the comparison between real and synthetic data, the diffusion model achieved inner-class and outer-class divergence values of 5.45769 and 0.02520, respectively, yielding the lowest ratio of 0.00462 among all methods. A lower ratio means a closer resemblance between synthetic and real data, suggesting that the diffusion model generates highly realistic synthetic signals. This substantial similarity makes diffusion-based synthetic data a valuable resource for training fall detection models, especially in scenarios where real sensor data are scarce. The synthetic data enhance model generalization and improve classification accuracy by supplementing the available real data. As a result, the diffusion model significantly reduces the reliance on extensive sensor data collection, thereby minimizing the associated costs in terms of workforce, time, and resources.

### 4.3. Computational Cost of the Data Augmentation Methods

The performance of fall detection systems can be enhanced through the application of data augmentation techniques. However, these methods often introduce substantial computational costs, resulting in longer preprocessing times and higher hardware requirements. As a result, careful consideration of computational efficiency is crucial when choosing appropriate data augmentation approaches.

Table 10 presents the computational costs of the evaluated augmentation methods, including both computation time and the number of model parameters. The computation time shown in Table 10 is measured from the beginning of the augmentation process through the completion of model training, encompassing the entire leave-one-group-out evaluation (with four groups in total).

Among the methods, the diffusion model exhibits the longest computation time due to its large number of parameters and the complexity of the forward and reverse diffusion processes. Although it demonstrates an outstanding ability to generate high-quality and diverse sensor data, the diffusion model requires the most substantial computational resources.

In contrast, traditional methods such as data transformation (DTF) and SMOTE require the shortest time to complete the augmentation and classification process. However, because these methods rely on simple statistical operations, the synthetic data they produce may lack fidelity and deviate from real sensor data, which could negatively impact system performance. Autoencoder and VAE demonstrate moderate computational demands. Autoencoders compress and reconstruct data through neural networks, while VAEs add complexity by learning probability distributions. While the data generated by these methods may not achieve the same level of realism as diffusion models, they offer a reasonable balance between computational efficiency and augmentation effectiveness, making them suitable alternatives when computational resources are limited.

### 4.4. The Influence of Real-to-Synthetic Data Ratio on Fall Detection Performance

Figure 9 illustrates the fall detection performance when trained with varying proportions of real and diffusion model-augmented synthetic data. The *x*-axis represents the ratio of synthetic data to real data. For example, if the total available real data are 100 training data, using 50% of the real data while generating 200% synthetic data means that 50 real samples are used to train the diffusion model, and a total of 150 samples (50 real and 100 synthetic) are used to train the fall detection model.

The results indicate that as the proportion of real data decreases, the F1 score of the fall detection model also declines. Regardless of the real-to-synthetic data ratio, applying diffusion model-based data augmentation consistently leads to better F1 scores compared with not using it. When the proportion of real data is at least 25%, the synthetic data effectively supplement the training set, achieving performance levels comparable with those obtained using an equivalent amount of real data. However, in cases where only 10% of the real data are available, the F1 score remains consistently low, around 73%, suggesting that the diffusion model’s effectiveness is still constrained by the quantity of training data in extremely low-data scenarios. Despite this limitation, the diffusion model demonstrates a strong capability to generate high-quality and diverse synthetic data, which can effectively serve as a substitute for real data, yielding results close to those achieved with real samples.

Figure 10 further highlights the improvement achieved by applying the diffusion model-based data augmentation compared with the baseline, which does not utilize augmentation. The enhancement is more pronounced when the proportion of real data is lower. Specifically, when only 25% of real data are used, generating 400% synthetic data improves the F1 score by 8.17%. In contrast, when training with 100% real data, the improvement is only 2.59% with the addition of 300% synthetic data. These findings underscore the substantial effectiveness of diffusion model-based data augmentation, particularly in scenarios where data are scarce. By reducing the need for large amounts of real data, this approach alleviates the burden of extensive data collection and annotation, making it a valuable tool for improving fall detection performance in resource-constrained environments.

## 5. Conclusions

Falls among the elderly pose a significant global health concern, making the development of a reliable fall detection system essential to mitigating the risks associated with fall-related injuries. A wrist-based fall detection system provides an ideal solution for the elderly due to its user-friendliness. However, the high complexity of hand motion modeling increases the difficulty of developing a reliable wrist-based fall detection system. In addition, collecting and annotating sensor data for fall detection is expensive and labor-intensive, leading to data scarcity and class imbalance issues. These challenges hinder the effectiveness of wrist-based fall detection systems using deep learning technology, as insufficient diverse training data limit model performance.

To address this problem, we compare several data augmentation methodologies, and a conditional diffusion model-based data augmentation method is recommended. By iteratively adding and removing noise from sensor signals, the diffusion model demonstrates a remarkable ability to generate high-quality and diverse synthetic data. Additionally, by incorporating class information into the generation process, this method effectively captures distinctive characteristics of each activity class, ensuring that the synthetic data closely align with the corresponding real activity patterns. As a result, the wrist-based fall detection model significantly enhances its ability to differentiate fall events from other activities accurately.

The results demonstrate that the conditional diffusion model-based data augmentation method outperforms other approaches, achieving the highest accuracy and F1 score in fall detection. When applied to the entire dataset, the method improves the F1 score by 2.59% compared with training without data augmentation. More notably, when the real training data are reduced to only 25%, the method still yields an F1 score improvement of 8.17%. These findings highlight the effectiveness of the diffusion model in compensating for data scarcity, providing a practical solution for maintaining fall detection system performance with limited real data. The success of this approach underscores its strong potential for real-world applications, further improving the reliability of wrist-based fall detection systems using deep learning technology.

## Figures and Tables

**Figure 1 sensors-25-02168-f001:**
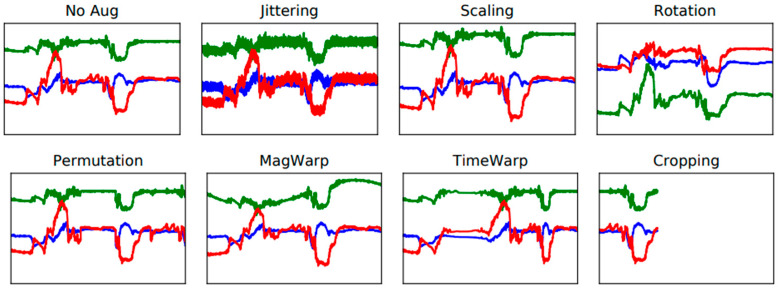
The illustration of various data transformation methods [30].

**Figure 2 sensors-25-02168-f002:**
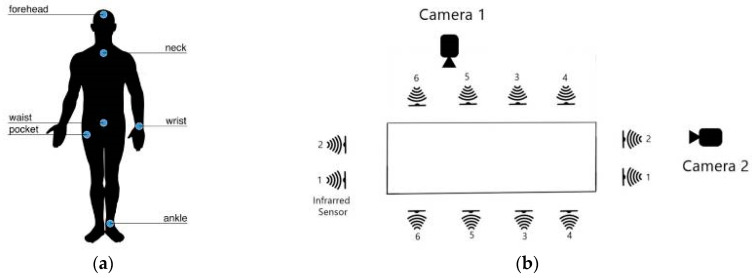
Distribution of the sensors. (**a**) Wearable sensors and EEG headset located on the human body. (**b**) Layout of the context-aware sensors and camera views [29].

**Figure 3 sensors-25-02168-f003:**
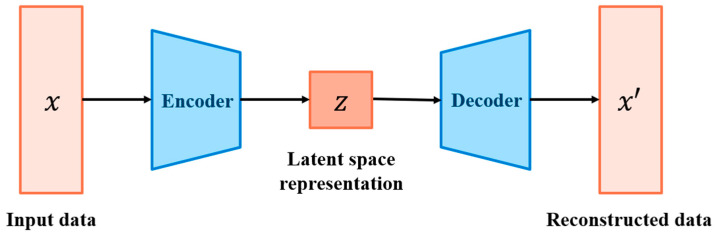
Autoencoder structure overview.

**Figure 4 sensors-25-02168-f004:**
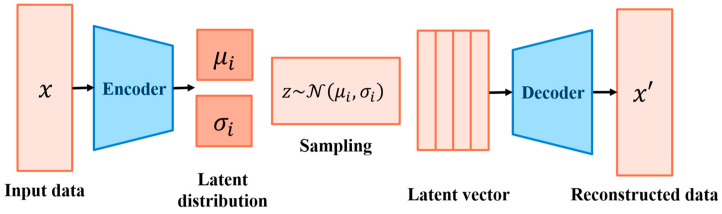
Variational autoencoder (VAE) structure overview.

**Figure 5 sensors-25-02168-f005:**
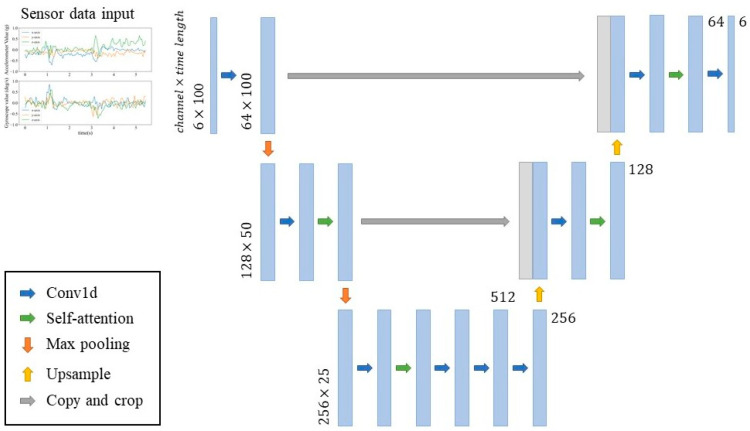
The architecture of the diffusion model.

**Figure 6 sensors-25-02168-f006:**
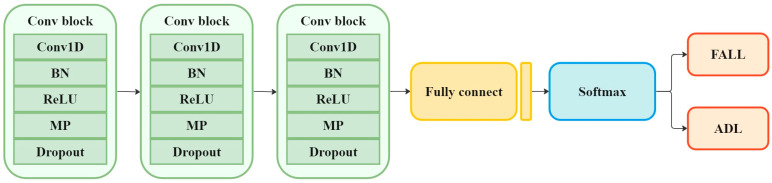
The architecture of fall detection system.

**Figure 7 sensors-25-02168-f007:**
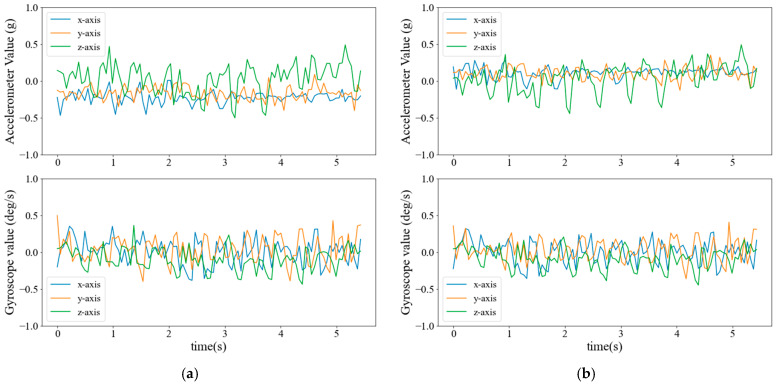
Accelerometer and gyroscope signals from the ADL events. (**a**) Real; (**b**) Synthetic.

**Figure 8 sensors-25-02168-f008:**
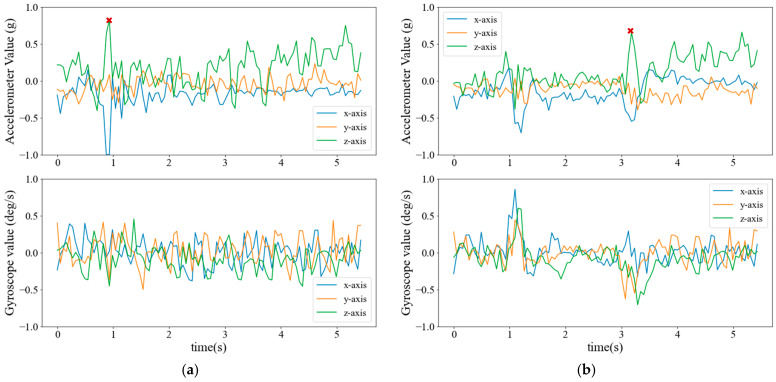
Accelerometer and gyroscope signals from the fall events. (**a**) Real; (**b**) Synthetic.

**Figure 9 sensors-25-02168-f009:**
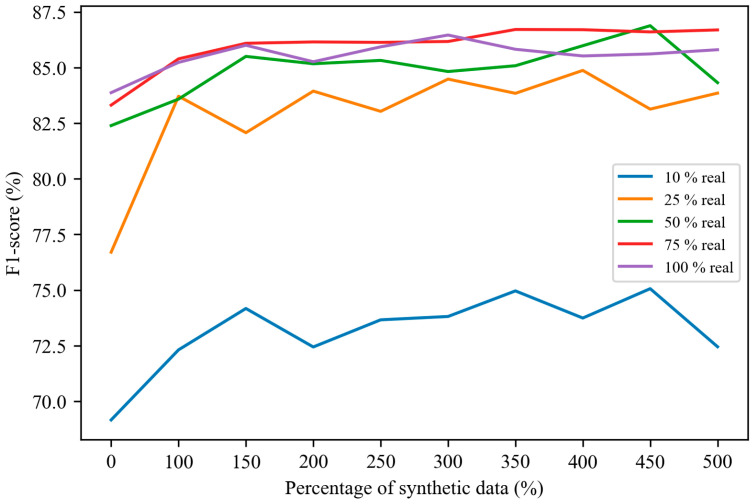
Fall detection performance with various ratio of real and diffusion model-augmented synthetic data.

**Figure 10 sensors-25-02168-f010:**
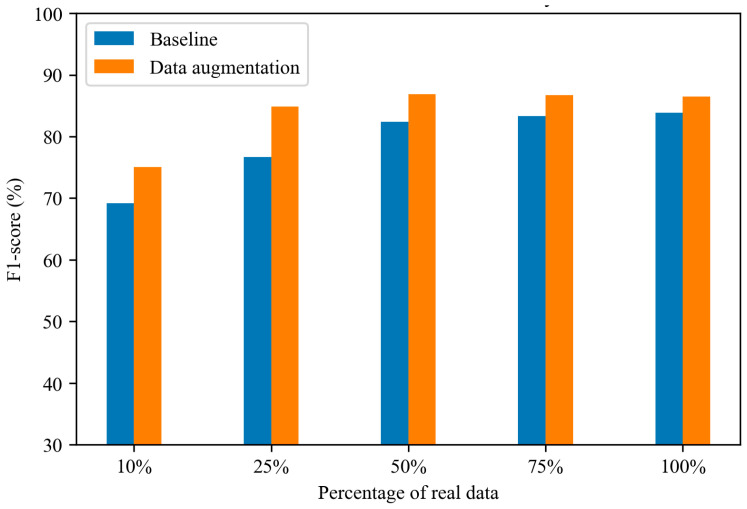
Optimal fall detection performance across different proportions of real and diffusion model-based synthetic data.

**Table 1 sensors-25-02168-t001:** Architecture of the autoencoder.

Module	Layer	Output_Size
**Encoder**	Conv1D (64, 3) MaxPool1D (2) Conv1D (128, 3) MaxPool1D (2) Conv1D (256, 3) MaxPool1D (2)	(64, 98) (64, 49) (128, 47) (128, 23) (256, 21) (256, 10)
**Decoder**	Upsample (28) Transposed Conv1D (128, 3) Upsample (61) Transposed Conv1D (64, 3) Upsample (126) Transposed Conv1D (6, 3)	(256, 21) (128, 23) (128, 47) (64, 49) (64, 98) (6, 100)

Conv1D—one-dimensional convolutional layer; MaxPool1D—one-dimensional max-pooling layer; Transposed Conv1D—one-dimensional transposed convolutional layer; Upsample—one-dimensional up-sampling layer.

**Table 2 sensors-25-02168-t002:** List of tried hyperparameters of autoencoder.

Hyperparameter	Searchspace
Learning rate	$Float\{10^{-5},10^{-4},3\times 10^{-4},5\times 10^{-4},10^{-3},5{\times 10}^{-3}\}$
Epochs	Int{50,100,300,500}
Conv1D layers	Int{2,3,4,5}
Conv1D (number of output channels)	Int{32,64,128,256,512}
Conv1D (kernel size)	Int{1,3,5}

**Table 3 sensors-25-02168-t003:** The architecture of the variational autoencoder (VAE).

Module	Layer	Output_Size
**Encoder**	Conv1D (64, 5) Conv1D (64, 5) Conv1D (64, 3) Conv1D (64, 3) MaxPool1D (2)	(64, 100) (64, 100) (64, 100) (64, 100) (64, 50)
**Bottleneck**	Linear (128)	(64, 128)
**Decoder**	Linear (50) Upsample (100) Conv1D (64, 3) Conv1D (64, 3) Conv1D (64, 5) Conv1D (6, 5)	(64, 50) (64, 100) (64, 100) (64, 100) (64, 100) (6, 100)

Conv1D—one-dimensional convolutional layer; MaxPool1D—one-dimensional max-pooling layer; Linear—linear transformation layer; Upsample—one-dimensional up-sampling layer.

**Table 4 sensors-25-02168-t004:** List of tried hyperparameters of VAE.

Hyperparameter	Searchspace
Learning rate	$Float\{10^{-5},10^{-4},3\times 10^{-4},5\times 10^{-4},10^{-3},5\times 10^{-3}\}$
Epochs	Int{50,100,300,500}
Conv1D layers	Int{2,3,4,5}
Conv1D (number of output channels)	Int{32,64,128,256,512}
Conv1D (kernel size)	Int{1,3,5}

**Table 5 sensors-25-02168-t005:** List of tried hyperparameters of diffusion model.

Hyperparameter	Searchspace
Learning rate	$Float\{10^{-5},10^{-4},3\times 10^{-4},5\times 10^{-4},10^{-3},5\times 10^{-3}\}$
Diffusion timesteps	Int{100,300,500,1000}
Epochs	Int{100,300,500}
Down and up-sampling layers	Int{2,3,4}

**Table 6 sensors-25-02168-t006:** Summary of the optimized hyperparameters for the autoencoder, VAE, and diffusion model.

Model	Hyperparameter	Value
Autoencoder	Learning rate	3 × 10^−4^
	Epochs	100
	Conv1D layers	3
	Conv1D (number of output channels)	64, 128, and 256, respectively, in three layers
	Conv1D (kernel size)	3
VAE	Learning rate	5 × 10^−4^
	Epochs	100
	Conv1D layers	4
	Conv1D (number of output channels)	64
	Conv1D (kernel size)	3 and 5, respectively, in four layers
Diffusion model	Learning rate	3 × 10^−4^
	Diffusion timesteps	1000
	Epochs	300
	Down and up-sampling layers	2

**Table 7 sensors-25-02168-t007:** Performance comparison of the fall detection system using different data augmentation methods (values in %).

Model	100% Real				25% Real			
	Accuracy	Precision	Recall	F1	Accuracy	Precision	Recall	F1
BL	92.94	91.64	78.59	83.87	88.08	75.26	79.97	76.70
DTF	92.93	90.88	79.66	84.14	90.73	80.30	83.83	81.45
SMOTE	92.89	89.09	81.08	84.44	90.39	77.35	86.42	81.44
AE	92.85	91.03	78.91	83.68	91.54	82.12	83.43	82.42
VAE	93.24	91.03	80.64	84.62	91.29	83.14	82.59	81.84
DM	93.30	85.79	87.05	86.00	91.75	82.17	86.07	83.28

**Table 8 sensors-25-02168-t008:** The comparison of TSTR score by different data augmentation methods (values in %).

Model	Accuracy	Precision	Recall	F1
DTF	99.64	98.92	99.64	99.28
SMOTE	96.69	89.39	98.41	93.67
AE	53.13	53.48	51.52	52.44
VAE	59.30	57.60	75.38	64.95
DM	91.30	75.51	96.45	84.67

**Table 9 sensors-25-02168-t009:** Divergence analysis of ADL and fall synthetic signals and the comparison between real and synthetic data.

Model	Fall/ADL			Real/Synthetic		
	Inner	Outer	Ratio	Inner	Outer	Ratio
DTF	0.0023	0.00003	0.0130	5.50319	0.04927	0.00895
SMOTE	0.0019	0.00006	0.0308	5.55463	0.04824	0.00868
AE	0.0018	0.00001	0.0056	5.70100	0.03291	0.00577
VAE	0.0003	0.000002	0.0062	5.43478	0.03999	0.00736
DM	0.0022	0.00009	0.0404	5.45769	0.02520	0.00462

**Table 10 sensors-25-02168-t010:** Comparison of computational costs for different data augmentation methods.

Model	Computation Time (s)	Parameter (Byte)
DTF	195.91	-
SMOTE	111.30	-
AE	250.81	207,128
VAE	283.76	462,144
DM	1165.32	21,115,928

## Data Availability

The original data presented in the study are openly available in UP-fall detection dataset at: https://sites.google.com/up.edu.mx/har-up/ (accessed on 25 February 2025).

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
