# Peer review of "Performance Analysis of Data Augmentation Approaches for Improving Wrist-Based Fall Detection System"

_sensors, 2025, doi:10.3390/s25072168_

Round 1
Reviewer 1 Report
Comments and Suggestions for Authors
I find the article interesting and generally clear and well written. It provides a study of the influence on performance of using synthetic data to detect falls with inertial sensors on the wrist, comparing different detection algorithms and proportion of synthetic data in the total data used for training. The article is technically correct in my opinion. Fundamentally I have comments on the form:
1.- It is convenient to revise the English. Examples: “is” instead of “are” in line 88, and “expended” instead of “expanded” in the same line (repeated in other places).
2.- Section 3.3.1: It is not clear whether the two signals, fall and daily activities, are augmented. The beginning of the section only refers to the fall signals, but in line 180 it refers to the daily activity signals.
3.- Line 181 states “the performance results obtained from the three transformations are averaged”. The tests are passed without distinguishing between synthetic data per transformation, if I am not mistaken. What do you mean by “averaged”?
4.- The index to refer to equation (7) is repeated (page 9).
5.- Please clarify the sentences between lines 309 and 311. Please clarify what B and W refer to in this paragraph. It is confusing because daily activity samples, fall samples, real data and synthetic data are mentioned.
6.- Explain better the paragraph between lines 312 and 317. A lower ratio than what ?
7.- Figure 6: I think it would be better understood if the changing signal is left alone or the signals are separated.
8.- Section 4.44: Table 4 instead of Table 1 (line 365).
9.- Figure 7: it would be convenient to point out the fall events in the figure.
10.- Section 4.3.2: When giving data (at least at the beginning) for inner-class divergence and outer-class divergence and their ratio, refer to the equations on page 9.
Comments on the Quality of English LanguageIt is convenient to revise the English. Examples: “is” instead of “are” in line 88, and “expended” instead of “expanded” in the same line (repeated in other places).
Author Response
Comments 1:
It is convenient to revise the English. Examples: “is” instead of “are” in line 88, and “expended” instead of “expanded” in the same line (repeated in other places).
Response 1:
Thank you for pointing this out. We acknowledge the errors in word choice and verb agreement, particularly in line 88, where “is” should be used instead of “are,” and “expanded” should be used instead of “expended.”
Comments 2:
Section 3.3.1: It is not clear whether the two signals, fall and daily activities, are augmented. The beginning of the section only refers to the fall signals, but in line 180 it refers to the daily activity signals.
Response 2:
We appreciate your feedback regarding the ambiguity in Section 3.3.1. We agree that our wording may have caused confusion about whether both fall and daily activity signals were augmented. To clarify this, we have revised the sentence to explicitly state, “augment both ADL and fall signals” in line 182. This modification ensures that readers clearly understand that both types of signals undergo augmentation.
Comments 3:
Line 181 states “the performance results obtained from the three transformations are averaged”. The tests are passed without distinguishing between synthetic data per transformation, if I am not mistaken. What do you mean by “averaged”?
Response 3:
Thank you for highlighting the lack of clarity regarding the term "averaged" in line 181. To address this, we have elaborated on the methodology by specifying that each transformation method is applied separately, and the resulting augmented datasets are individually evaluated. The final performance results are then averaged to assess their overall effectiveness. The revised text, now located between lines 188 and 191, reads:
“These transformations are applied separately to generate augmented versions of the signals and are individually evaluated to assess their impact on the fall detection system. Finally, the results obtained from the three different transformation methods are averaged to determine their overall effectiveness in enhancing data augmentation.”
Comments 4:
The index to refer to equation (7) is repeated (page 9).
Response 4:
We appreciate your careful review and for pointing out the repeated equation reference on page 10. We have corrected this mistake by updating the incorrect reference from equation (7) to equation (8). Thank you for your attention to detail.
Comments 5:
Please clarify the sentences between lines 309 and 311. Please clarify what B and W refer to in this paragraph. It is confusing because daily activity samples, fall samples, real data and synthetic data are mentioned.
Response 5:
Thank you for your comment regarding the unclear explanation of "B" and "W" in lines 343 to 344. We acknowledge that the previous wording may have caused confusion, especially with multiple references to daily activity samples, fall samples, real data, and synthetic data. The terms "B" and "W" originate from the cited study and are used as abbreviations for inter-class dispersion (S_B) and intra-class dispersion (S_W). Since these terms are consistent with the original study, we have retained them while improving the explanation for better clarity.
To enhance readability, we have restructured this section (lines 345–351) to separately describe the two key relationships being analyzed:
The distinction between fall and ADL signals within the synthetic dataset. A higher ratio in this context indicates greater class separability, suggesting improved differentiation between ADL and fall signals.
The similarity between real and synthetic data. Here, an equal number of samples from each class (both real and synthetic) are randomly selected. A lower ratio in this case suggests that the synthetic data closely resembles the real data, reflecting higher realism in the generated samples.
By separating these explanations, we aim to make the discussion more intuitive and reduce potential confusion.
Comments 6:
Explain better the paragraph between lines 312 and 317. A lower ratio than what?
Response 6:
Thank you for your suggestion regarding the paragraph between lines 312 and 317. We acknowledge that the meaning of "a lower ratio" was unclear. This issue has been addressed in our response to Comment 5, where we now explicitly clarify that the lower ratio refers to the similarity between real and synthetic data. A lower value in this context indicates that the synthetic data closely resembles real data, thereby demonstrating higher realism. This explanation has been incorporated into lines 345–351 to ensure a clearer understanding.
Comments 7:
Figure 6: I think it would be better understood if the changing signal is left alone or the signals are separated.
Response 7:
Thank you for your valuable suggestion. We have made several modifications to improve the clarity of Figure 7 and Figure 8 (originally Figure 6 and Figure 7). Specifically, we have added the unit labels on the y-axis and corrected the x-axis numbering to ensure accurate representation of the data. Regarding the presentation of the three-axis signals, we have chosen to maintain the current format. The reason for this decision is that the synthetic signals generated by the diffusion model are not directly mapped to a specific real signal. Instead, they represent a learned distribution of the real data rather than an exact one-to-one correspondence. Separating the signals or isolating a single changing signal may not fully capture the intended comparison between real and synthetic signals. However, we acknowledge the importance of clear visualization and will consider additional adjustments if needed to enhance readability and interpretation. Thank you again for your insightful feedback.
Comments 8:
Section 4.44: Table 4 instead of Table 1 (line 365).
Response 8:
Thank you for pointing this out. We have corrected the reference in line 364, changing "Table 1" to "Table 7" to ensure accuracy.
Comments 9:
Figure 7: it would be convenient to point out the fall events in the figure.
Response 9:
We appreciate this suggestion. To enhance clarity, we have revised Figure 8 (originally Figure 7) by adding an “x” mark to indicate the critical points of the fall events. This modification helps better illustrate the characteristic peak associated with fall events. The updated figure now explicitly highlights these key moments. Additionally, we have revised the corresponding description to state in line 402-403:
“To illustrate this characteristic, the highest acceleration value in Figure 8 is marked with an ‘x.’”
Comments 10:
Section 4.3.2: When giving data (at least at the beginning) for inner-class divergence and outer-class divergence and their ratio, refer to the equations on page 9.
Response 10:
We sincerely apologize for this oversight. The issue arose because the outer-class divergence value is extremely small, leading to a significant discrepancy between the actual value and the one displayed in the table due to omitted decimal places. To correct this, we have recalculated the ratio based on the values shown in the Table 9 and have ensured that the correct ratio is now presented accurately.

Reviewer 2 Report
Comments and Suggestions for Authors
- Re-draw figure 1. It is too small to read. 2*4 or 4*2 pattern is more acceptable.
- Figure 2 and 3 are useless. We could not understand the detail information, such as how to encode, or how to decoder.
- Author improves the F1-score by 6.58% with only 25% of the actual data. However, a comparison between the present research and other’s should be carried out to demonstrate the highlight of the present work.
- There are no figures to show the equipment/sensors/systems the present study used. MDPI SENSORS concern the fabrication and application of various sensors. This work mainly concern the data processing (fall detection system) by deep learning.
- There is no on-body test or field test of the developed fall detection system. Further supplement study is recommended to carried out.
Author Response
Comments 1:
Re-draw figure 1. It is too small to read. 2*4 or 4*2 pattern is more acceptable.
Response 1:
Thank you for your suggestion. We have re-drawn Figure 1 in a 4×2 layout to improve its readability. This adjustment ensures that the figure is clearer and easier for readers to interpret.
Comments 2:
Figure 2 and 3 are useless. We could not understand the detail information, such as how to encode, or how to decode.
Response 2:
We appreciate your feedback regarding Figures 2 and 3. To enhance their clarity and usefulness, we have expanded the accompanying textual descriptions to provide a more detailed explanation of the mechanisms behind the autoencoder and VAE models.
For the autoencoder, we have included the following descriptions to clarify its structure and functionality:
- "The autoencoder, composed of an encoder and a decoder, is illustrated in Figure 3. The encoder compresses input signals into a latent space representation that preserves essential information."
- "The decoder’s objective is to reconstruct the original signals while retaining key features from the encoded representation."
These explanations are now incorporated between lines 201 and 202, and between lines 206 and 207 to improve reader comprehension.
For the VAE, we have provided additional details on how it encodes and decodes information:
- "In this framework, the encoder compresses input data into a latent distribution, learning to represent it using a mean (μ) and standard deviation (σ)."
- "At the bottleneck, two fully connected layers transform the latent features into the mean and standard deviation, defining a probabilistic latent space. The decoder starts with an upsampling layer of size 100, followed by four convolutional layers that reconstruct the data from the latent representation."
These clarifications have been added between lines 228 and 229, and between lines 231 and 234 to better explain the concepts presented in Figures 4.
We hope these revisions make the figures more informative and enhance the overall understanding of our methodology. Thank you again for your valuable input.
Comments 3:
Author improves the F1-score by 6.58% with only 25% of the actual data. However, a comparison between the present research and other’s should be carried out to demonstrate the highlight of the present work.
Response 3:
Thank you for your valuable feedback regarding the comparison with other researches. As this study is, to the best of our knowledge, the first to explore data augmentation methods in fall detection systems, specifically focusing on diffusion models, we are unable to directly reference similar works. Additionally, due to variations in datasets and data preprocessing techniques, direct comparisons with other studies are challenging. However, we believe that this paper can make a meaningful contribution by providing a comparison of different data augmentation techniques, discussing their impact on the fall detection system, as well as examining the quality of synthetic data and the real-to-synthetic data ratio.
Comments 4:
There are no figures to show the equipment/sensors/systems the present study used. MDPI SENSORS concern the fabrication and application of various sensors. This work mainly concerns the data processing (fall detection system) by deep learning.
Response 4:
Thank you for your valuable suggestion. We acknowledge the importance of illustrating the equipment, sensors, and systems used in our study. To address this, we have added Figure 2, which is cited from the UPFall public dataset. This figure illustrates the distribution of the sensors, including both the wearable sensor placements and the fixed sensor locations, offering readers a clearer understanding of how the data was collected.
Comments 5:
There is no on-body test or field test of the developed fall detection system. Further supplement study is recommended to carried out.
Response 5:
We appreciate your feedback regarding the lack of an on-body test or field test in our study.
As our research utilizes a publicly available dataset, we selected this approach due to its ease of access and fairness, ensuring that our proposed method can be objectively evaluated and compared with existing studies. However, we recognize the importance of real-world validation. To further enhance the credibility and practical applicability of our fall detection system, we plan to conduct on-body tests in next step. This will allow us to validate our proposed methods under real-world conditions and assess their effectiveness in practical applications.

Reviewer 3 Report
Comments and Suggestions for Authors
This manuscript presents various data augmentation methods to enhance the performance of a fall detection system based on wrist joints using deep learning techniques. The conditional diffusion model proves to be an effective data augmentation approach, showing a 6.58% increase in f1 score when trained on only 25% of actual data, indicating substantial potential. The reviewers provided specific feedback as follows:
- When considering the comparison of different methods in the study, it is important to take into account the issue of computational resources for fall detection systems. In cases where computational resources are limited, such as in embedded systems, traditional data transformation and SMOTE may be more suitable, although manual parameter tuning is required to reduce data distortion. On the other hand, if the goal is to generate high-quality fall samples, VAE and DDPM show more potential, albeit with higher computational costs.
- Due to the differing window partitioning methods between the ADL and fall datasets, the model may learn variations in handling rather than actual behavioral differences during training. Employing leave-one-subject-out cross-validation (LOSO-CV) ensures the model generalizes across different individuals..
- Figures 6 and 7 recommend that the format of images be standardized and that X\Y\Z-axis be added to improve readability.
Author Response
Comments 1:
When considering the comparison of different methods in the study, it is important to take into account the issue of computational resources for fall detection systems. In cases where computational resources are limited, such as in embedded systems, traditional data transformation and SMOTE may be more suitable, although manual parameter tuning is required to reduce data distortion. On the other hand, if the goal is to generate high-quality fall samples, VAE and DDPM show more potential, albeit with higher computational costs.
Response 1:
Thanks for the constructive suggestions. We totally agree with this comment. Therefore, we have deeply discussed the computational cost of different data augmentation approaches and expended our article at section 4.3. (line 452-479), and the statements are as follow:
4.3. Computational cost of the data augmentation methods
The performance of fall detection systems can be enhanced through the application of data augmentation techniques. However, these methods often introduce substantial computational, resulting in longer preprocessing times and higher hardware requirements. As a result, careful consideration of computational efficiency is crucial when choosing appropriate data augmentation approaches.
Table 7 presents the computational costs of the evaluated augmentation methods, including both computation time and the number of model parameters. The computation time shown in Table 7 is measured from the beginning of the augmentation process through the completion of model training, encompassing the entire leave-one-group-out evaluation (with four groups in total).
Among the methods, the diffusion model exhibits the longest computation time due to its large number of parameters and the complexity of the forward and reverse diffusion processes. Although it demonstrates an outstanding ability to generate high-quality and diverse sensor data, the diffusion model requires the most substantial computational resources.
In contrast, traditional methods such as data transformation (DTF) and SMOTE require the shortest time to complete the augmentation and classification process. However, because these methods rely on simple statistical operations, the synthetic data they produce may lack fidelity and deviate from real sensor data, which could negatively impact system performance. Autoencoder and VAE demonstrate moderate computational demands. Autoencoders compress and reconstruct data through neural networks, while VAEs add complexity by learning probability distributions. While the data generated by these methods may not achieve the same level of realism as diffusion models, they offer a reasonable balance between computational efficiency and augmentation effectiveness, making them suitable alternatives when computational resources are limited.
Comments 2:
Due to the differing window partitioning methods between the ADL and fall datasets, the model may learn variations in handling rather than actual behavioral differences during training. Employing leave-one-subject-out cross-validation (LOSO-CV) ensures the model generalizes across different individuals.
Response 2:
Thanks for the reviewer’s comment. In this study, a similar cross-validation approach, the leave-one-group-out cross-validation (LOGO-CV) method, was employed to ensure the model generalizes across different individuals and described in lines 265-271: “This validation approach simulates a realistic scenario where the model is tested on subjects whose data are not included in the training set. The dataset, consisting of 17 subjects, is divided into four groups: three groups containing four subjects each and one group containing five subjects. During training, data from three groups are used to train the model, while the remaining group is held out for testing. This process is repeated until all groups have been used for testing, ensuring a robust evaluation of the model’s performance across different subjects.”
Comments 3:
Figures 6 and 7 recommend that the format of images be standardized and that X\Y\Z-axis be added to improve readability.
Response 3:
Thanks for the reviewer’s recommendation. The improved figures are revised with index of Figure 7 and Figure 8.

Reviewer 4 Report
Comments and Suggestions for Authors
This manuscript presents a comprehensive performance analysis of various data augmentation methods for enhancing wrist-based fall detection systems using deep learning technology. The study addresses a significant issue in the field of elderly healthcare and provides valuable insights into improving the reliability of fall detection systems. The research is well-structured, and the results are promising. However, there are some areas that could be improved to strengthen the manuscript.
1, While the data preprocessing section is thorough, more details could be provided regarding the handling of missing data or outliers in the dataset. Additionally, a sensitivity analysis of the preprocessing steps on the final results would strengthen the study.
2, The manuscript mentions the use of Adam optimizer with specific learning rates for different models. However, a discussion on the selection of these hyperparameters and their impact on model performance would be beneficial. Including a hyperparameter optimization process could further validate the results.
3, The results section could benefit from a more in-depth analysis of why certain data augmentation methods perform better than others. For instance, exploring the specific characteristics of the diffusion model that make it superior in generating synthetic data could provide deeper insights.
4, The study focuses on improving fall detection performance using synthetic data. However, a discussion on the practical aspects of deploying such a system in real-world scenarios, including computational efficiency and hardware requirements, would be valuable for readers interested in implementing the system.
5, The references to relevant literature in the manuscript are not comprehensive enough. It is recommended to include more related references (eg., 10.34133/research.0154,10.1002/EXP.20230046,10.1002/adfm.202412307) to enrich the article.
Author Response
Comments 1:
While the data preprocessing section is thorough, more details could be provided regarding the handling of missing data or outliers in the dataset. Additionally, a sensitivity analysis of the preprocessing steps on the final results would strengthen the study.
Response 1:
Thanks for reviewer’s comment. We totally agree with this comment. In this study, we employed a public dataset, UP-Fall Detection Datasets, to validate different data augmentation methodologies. Firstly, the missing data are removed at the start of preprocessing stage, where we had revised our article at line 159-160. As for the issue of outliers, it is essential to proposed a method carrying a great generalization ability. Hence, the outliers won’t be screen in this study to ensure the ability of generalization for each data augmentation approaches.
Comments 2:
The manuscript mentions the use of Adam optimizer with specific learning rates for different models. However, a discussion on the selection of these hyperparameters and their impact on model performance would be beneficial. Including a hyperparameter optimization process could further validate the results.
Response 2:
Thanks for the constructive suggestions. In this study, different models have their specific hyperparameter should be tuned, which we had revised and listed all parameters at Table2 for autoencoder, Table 4 for VAE, and Table 5 for diffusion model. All the hyperparameter have been tuned and optimized. Therefore, we list the main hyperparameters and the optimized values, which we recommended using, in Table 6, and the following results are performed based on the optimized hyperparameters.
Comments 3:
The results section could benefit from a more in-depth analysis of why certain data augmentation methods perform better than others. For instance, exploring the specific characteristics of the diffusion model that make it superior in generating synthetic data could provide deeper insights.
Response 3:
Thanks for the reviewer’s comment. We admit that description of the notable characteristics of the diffusion model is not clear enough. Therefore, we have added more deeper discussion for computational cost of different data augmentation methods at section 4.3. in line 441-467, and the statements are as follow:
4.3. Computational cost of the data augmentation methods
The performance of fall detection systems can be enhanced through the application of data augmentation techniques. However, these methods often introduce substantial computational, resulting in longer preprocessing times and higher hardware requirements. As a result, careful consideration of computational efficiency is crucial when choosing appropriate data augmentation approaches.
Table 7 presents the computational costs of the evaluated augmentation methods, including both computation time and the number of model parameters. The computation time shown in Table 7 is measured from the beginning of the augmentation process through the completion of model training, encompassing the entire leave-one-group-out evaluation (with four groups in total).
Among the methods, the diffusion model exhibits the longest computation time due to its large number of parameters and the complexity of the forward and reverse diffusion processes. Although it demonstrates an outstanding ability to generate high-quality and diverse sensor data, the diffusion model requires the most substantial computational resources.
In contrast, traditional methods such as data transformation (DTF) and SMOTE require the shortest time to complete the augmentation and classification process. However, because these methods rely on simple statistical operations, the synthetic data they produce may lack fidelity and deviate from real sensor data, which could negatively impact system performance. Autoencoder and VAE demonstrate moderate computational demands. Autoencoders compress and reconstruct data through neural networks, while VAEs add complexity by learning probability distributions. While the data generated by these methods may not achieve the same level of realism as diffusion models, they offer a reasonable balance between computational efficiency and augmentation effectiveness, making them suitable alternatives when computational resources are limited.
In addition, we have also revised the description of the results presented in Table 7 (lines 378–385) as follows:
“This notable improvement can be attributed to the ability of diffusion models to gen-erate high-quality, diverse synthetic samples. By leveraging probabilistic modeling, diffu-sion models learn the underlying data distribution. Through an iterative process of pro-gressively adding and removing noise, they generate diverse and realistic synthetic sam-ples that enhance the training set. This diversity enables the classifier to learn more gener-alizable and discriminative features, leading to improved performance, especially when real data is limited. These results highlight the strong potential of diffusion models for augmenting sensor data in fall detection applications.”
Comment 4:
The study focuses on improving fall detection performance using synthetic data. However, a discussion on the practical aspects of deploying such a system in real-world scenarios, including computational efficiency and hardware requirements, would be valuable for readers interested in implementing the system.
Response 4:
Thanks for the constructive suggestions. We totally agree with this comment. However, we aim to provide a general solution of data augmentation for wrist-based fall detection system in this study. Through implementing various data augmentation methodologies, the qualitative analysis results such as performance enhancement and the quality of synthetic data are evaluated. Therefore, the further analysis will be immediately accessed on our next stage of our study to deeply figure out the real-world situation. In addition, the discussion of computational cost is added to the article at section 4.3. in line 452-479.
Comments 5:
The references to relevant literature in the manuscript are not comprehensive enough. It is recommended to include more related references (eg., 10.34133/research.0154,10.1002/EXP.20230046,10.1002/adfm.202412307) to enrich the article.
Response 5:
Thanks for the reviewer’s comment. Agree. We have, accordingly, added some references to emphasize this point with reference index of 13 and 14 in line 49.

Round 2
Reviewer 2 Report
Comments and Suggestions for Authors
Authors have addressed my proposed issues.